# Antibiotic Consumption, Illness, and Maternal Sensitivity in Infants with a Disorganized Attachment

**DOI:** 10.3390/children10071232

**Published:** 2023-07-17

**Authors:** Marina Fuertes, Anabela Faria, Joana L. Gonçalves, Sandra Antunes, Francisco Dionisio

**Affiliations:** 1Centro de Psicologia, University of Porto, 4200-135 Porto, Portugal; marinaf@eselx.ipl.pt; 2Escola Superior de Educação de Lisboa, Instituto Politécnico de Lisboa, 1549-020 Lisboa, Portugal; 3Hospital de Santo Espírito da Ilha Terceira, 9700-049 Angra do Heroísmo, Portugal; anabela.rb.faria@azores.gov.pt; 4Instituto de Psicologia e Ciências da Educação, Universidade Lusíada de Lisboa, 1349-001 Lisboa, Portugal; joanaglopes@gmail.com; 5inED–Centro de Investigação e Inovação em Educação, Escola Superior de Educação, Instituto Politécnico do Porto, 4200-465 Porto, Portugal; 6Department of Social and Behavior Sciences, University of Maia—ISMAI, 4475-690 Maia, Portugal; 7Faculdade de Psicologia, Universidade de Lisboa, 1649-013 Lisboa, Portugal; sandra.antunes@estimulopraxis.com; 8cE3c—Centre for Ecology, Evolution and Environmental Changes & CHANGE—Global Change and Sustainability Institute, Faculdade de Ciências, Universidade de Lisboa, 1749-016 Lisboa, Portugal

**Keywords:** mother–infant attachment, disorganized attachment, maternal sensitivity, infant cooperation, antibiotic uptake, prematurity

## Abstract

Prior research found an association between mother–infant attachment and antibiotic use. Ambivalent-attached infants are more likely to take antibiotics than other infants, and their mothers tend to be less sensitive to their needs than most. This finding is important because it shows the association between psychological processes, early relationships, and health outcomes. We aim to learn about children with high-risk attachment relationships, such as disorganized-attached infants. This study compares antibiotic use, infant–mother interactive behavior, and health indicators according to infant attachment patterns (including disorganized attachment). For this purpose, we observed mothers–infants’ interactive behavior in free play at nine months and infants’ attachment in the Ainsworth Strange Situation at twelve months. Participants included 77 girls and 104 boys (full-term and preterm) and their mothers. Paradoxically, mothers of disorganized-attached infants reported that their children were ill only 1.56 times on average, but 61% of their children used antibiotics in the first nine months. The other mothers reported that their children were sick 5.73 times on average, but only 54% of their children used antibiotics in the same period. Infants with disorganized attachment had mothers who were more literate and less sensitive. These results add to a body of research that shows that early high-risk relationships affect children’s lives at multiple levels.

## 1. Introduction

Recent research has found an association between maternal behavior, infant attachment, and antibiotic use. Fuertes et al. studied infants born at full term and found that the proportion of ambivalent-attached infants that took antibiotics in the first nine months of life was significantly higher than among secure and avoidant-attached infants [1]. Further studies found that the mothers of infants who consumed antibiotics were more anxious and less sensitive than those whose children used antibiotics [2]. One possible explanation relies on the higher levels of externalization and expression of negativity of ambivalent-attached infants when ill. In response, attachment figures tend to become worried and look for medical treatment and ways to relieve the infants’ suffering, expressing their concerns to medical professionals, which may lead to antibiotic use as a preventive measure. Nevertheless, the increased likelihood of ambivalent-attached infants taking antibiotics is not observed among infants with low gestational weight. Likely, early intervention and health services supported some of these infants and their families [3]. In sum, the risk associated with relationships seems to impact antibiotic use more than biological vulnerabilities such as prematurity or low gestational weight [4]. The present study investigates the association between antibiotic use and maternal sensitivity in infants at high risk for attachment relationships. We focus on disorganized-attached infants and compare them with secure and insecure attachments.

### 1.1. But Why Is It Essential to Study the Factors Associated with Antibiotic Use during Childhood?

It is undeniable that antibiotics represent an enormous medical discovery of the 20th century. However, bacteria frequently become antibiotic-resistant. There are two main routes toward resistance.

First, in some instances, specific single mutations in the bacterial chromosome are sufficient to turn antibiotic-sensitive bacterial cells into antibiotic-resistant. These are the cases, for example, of antibiotics as different as rifampicin (which acts on RNA synthesis), streptomycin (which inhibits protein synthesis), or nalidixic acid (which blocks DNA replication, and hence cell division). These mutations may appear within a few hours of bacterial growth. In the presence of antibiotics, sensitive bacteria die or halt cell division, leaving resources to antibiotic-resistant cells [5]. 

The second route toward antibiotic resistance is based on a phenomenon common among bacteria but rare or absent in other living beings: bacteria often receive DNA from the environment and other bacterial cells. These moving DNA strands (e.g., plasmids) frequently encode genes that confer antibiotic resistance [5]. These DNA elements may originate from non-pathogenic cells; therefore, even non-pathogenic bacteria may contribute to the spread of drug resistance [6]. Moreover, sensitive pathogenic bacteria may tolerate or persist despite the antibiotics’ presence [7]. Furthermore, because resistant cells often survive antibiotics thanks to the production of antibiotic-degrading enzymes, drug-sensitive pathogenic bacteria may thrive under antibiotics with the help of nearby resistant cells. Resistant and sensitive cells may belong to the same or different species [8,9,10]. 

These Darwinian processes, together with the massive use of antibiotics worldwide for decades, have had negative consequences: the levels of antibiotic resistance among pathogens have increased a lot [11,12]. Murray and colleagues recently showed that more than a million people died worldwide in 2019 due to antibiotic resistance [13]. As Laxminarayan writes in a companion commentary [14], this number of deaths is similar to the sum of HIV-related and malaria deaths [15,16]. The study by Murray et al. also estimated that about 5 million deaths in 2019 were associated with antibiotic resistance [13]. In sum, we live in a pandemic of antibiotic resistance [14,17,18]. Other recent studies have also reached similar worrisome conclusions [19,20]. To give some perspective on this problem, we further mention that the number of publications in the Scopus database referring to “antibiotic resistance” in the title, abstract, or keywords increased much more than the database itself. While “antibiotic resistance” increased 19-fold between 1983 and 2022 (from 7.78 × 10^2^ publications/year in 1983 to 4.70 × 10^4^ publications/year in 2022), the database only increased 8-fold in the same period (from 4.32 × 10^5^ publications/year in 1983 to 3.47 × 10^6^ publications/year in 2022). World Health Organization considers antibiotic resistance as one of the biggest global public health threats facing humanity [21,22].

Antibiotic use has another downside: these drugs can be detrimental at the individual level, most likely due to the perturbation of the microbiota colonization process. For example, in infants, there is an association between antibiotic use and the development of obesity, allergies, and type 1 diabetes [23,24,25]. Preterm, particularly very-to-extreme preterm infants, are even more susceptible to these drugs than full-term infants [26,27,28,29]. Antibiotics (mainly broad-spectrum antibiotics) can interfere with brain development [30,31,32], for example, increasing the risk of severe mental disorders later on as adults [33,34]. Moreover, Slykerman et al.’s research has linked the use of antibiotics in the first year of life to later neurocognitive impacts in children, such as anxiety and depression [35].

### 1.2. Infant Attachment Strategy and Antibiotic Consumption

In 1969, Bowlby postulated that humans have an attachment system that allows them to bond to their attachment figures. With this evolutionary-selected emotional tie, based on a motivational system, children obtain proximity, protection, and comfort from their attachment figures when feeling pain, fatigue, or sickness [36]. The attachment system provides an evolutionary advantage: children are more likely to survive adulthood with the attachment figures’ protection, comfort, and care [36]. 

A few years later, Ainsworth et al. developed and used an experimental paradigm (Strange Situation) designed to elicit attachment strategies [37], describing three patterns of attachment: secure (B), insecure-avoidant (A), and insecure-ambivalent (C) [38]. According to Ainsworth et al., “infants develop a secure attachment when the caregiver is sensitive to their signals and responds appropriately to their needs”. Infants with a secure attachment trust their attachment figures and expect to be comforted in distress. Therefore, infants with a secure attachment are easily soothed by these figures. Without a secure base, avoidantly attached infants avoid contact with the attachment figure when distressed; their attachment figure is frequently invasive, too demanding, or punitive, reinforcing the infants’ distress. Last, infants with an ambivalent attachment show inconsistent proximity seeking while declining contact with their attachment figure (*wanted but not wanted*). These infants are difficult to soothe and comfort by their attachment figure [38]. In turn, their caregivers are often inconsistent, overprotective, or absent. Both partners seem trapped in an unresolved interaction [37,38].

Mother–infant patterns of attachment impact children’s emotional, linguistic, and cognitive development, as well as self-regulation and health outcomes [39,40,41,42]. Interestingly, children with chronic illnesses tend to establish an insecure attachment with their mothers more than children with a secure attachment [42,43,44,45]. However, individuals with an insecure-avoidant versus an insecure-ambivalent attachment pattern respond differently to pain and sickness [42,42,46], meaning there are differences between the two insecure attachment styles. With the same symptoms, insecure-ambivalent individuals display high levels of emotionality, whereas avoidant-attached children and teenagers exhibit low levels of negativity and high levels of emotional control [41].

### 1.3. High-Risk Attachment Relationships and Different Views on Atypical Attachment

In the Strange Situation, a small group of infants displays conflicted, disoriented, or fearful behaviors on reunions with their caregivers. Ainsworth and her team found these infants could not be placed in the original attachment classification of secure, insecure-avoidant, and insecure-ambivalent-resistant patterns [38]. Later, researchers provided several classifications and descriptions to describe the behavior of these infants. Arguably, the disorganized (D) description is the most popular classification [47,48]. Main and Solomon detected certain behavior types that can be considered disorganized or disoriented, namely: (1) sequential and (2) simultaneous display of contradictory behavior patterns; (3) undirected, misdirected, incomplete, and interrupted movements and expressions; (4) stereotypies, asymmetrical, and mistimed movements and anomalous postures; (5) freezing, stilling, and slowed movements and expressions; (6) direct indices of apprehension regarding the parent; and (7) direct indices of disorganization and disorientation [47,48].

It is the intensity of the display of conflict, disorientation, or fear and the extent to which it is incorporated into the children’s attachment strategy that leads to a disorganized attachment classification. For this reason, disorganized attachment is an additional classification to the Ainsworth original one. Thus, coders must code the cases with the original classification and use the second classification when the infant’s behavior does not fit the original categories.

However, not all attachment researchers accept the concept of disorganization. The term “disorganized” can be misunderstood as random. Crittenden (2006) has argued that human evolution does not favor random non-self-protective behaviors since infants need their caregiver’s protection and care during the first years of life to survive [49]. Thus, robust mother–infant bonding must be in place to protect the offspring and ensure its survival until the fertile age. According to Crittenden (1999), “danger creates the need and the occasion for humans to develop the capacity to organize” [50]. Today’s humans descend from other humans and primates that must have survived several dangers, including predators, child maltreatment, wars, and natural disasters. Therefore, humans must have evolved strategies to face threats.

There are other atypical styles of attachment already described. Crittenden (1992) described the Avoidant/Resistant (A/C) pattern [51]. Infants with this strategy exhibit moderate to high avoidance and moderate to high resistance during reunions, combined with moderate to high proximity seeking and contact maintenance. For Crittenden (1999), the secure pattern is not the “best” strategy—it is just one of several attachment strategies that result from the “environment of evolutionary adaptedness” [50]. Each strategy is adaptive to their rearing context; for instance, the A/C pattern allows infants to “flight or fight” according to the changes in the context. That strategy can be efficient with unpredictable, scary and/or unstable parents [52]. On the other hand, Lyons-Ruth described the Unstable-Avoidant (UA) pattern [53]. These infants show a marked avoidance in the first reunion in the Strange Situation paradigm, followed by a conspicuous drop in avoidance in the second reunion. Overall, these attachment patterns are rare compared to the traditional classification and correspond to an unbalanced attachment strategy. Our study uses the D classification because it is more common than A/C or UA, increasing the statistical power of analyses. 

The causes for attachment quality are diverse, multisystemic, and mutually affected. However, past research found that frightening, frightened, and dissociative caregiver behaviors are associated with elevated rates of infant disorganized attachment [54,55,56].

Also, parents with unresolved losses, depression, or mental problems are more likely to exhibit these disruptive caregiving behaviors [57]. It is important to stress that there is no linear association between caregiving and disorganized attachment—instead, it is a probabilistic association affected by multiple factors [58]. In light of this perspective, one must consider the evidence that children who are victims of maltreatment or neglected parenting have high rates of disorganized attachment compared to children with low-risk backgrounds [59]. Regardless of the causes, disorganized attachment is a low-moderate predictor for developing social and behavioral problems and mental health problems in childhood and adolescence [60,61].

Most studies have used the Ainsworth original scales because the disorganized/disoriented pattern is rare (generally less than 5% of the normative samples) and more likely in at-risk samples. However, it is essential to learn about the behavior and relationships of these children at risk for socio-emotional development and use this knowledge to propose preventive interventions. We aim to study the associations between infant and maternal interactive behavior, health factors, and infant attachment patterns.

### 1.4. Aims

There are three main goals for the current investigation. First, we compare infants’ attachment strategies (securely attached, avoidantly attached, ambivalent attached, or disorganized) with antibiotic usage. To our knowledge, this is the first study comparing all four attachment groups concerning antibiotic use. Additionally, we aim to compare maternal sensitivity and maternal factors in all groups (e.g., maternal age, maternal education). Last, we aim to study the association between antibiotic consumption and infant and maternal interactive behavior in each group. Because normative low-risk samples generally have little or no cases of disorganized attachment, in this sample, we included infants born preterm since many of their mothers reported early maternal experiences as traumatic [62], and past research found a higher incidence of disorganized attachment in this group compared to samples with infants born full-term [63,64].

## 2. Materials and Methods

### 2.1. Participants

This sample is a subset sample of a previous study with 188 mother–infant dyads published [4]. In that study, seven infants had low birthweight to their gestational age. These cases were excluded to decrease the heterogeneity of the sample. Therefore, the final sample comprised 181 infants (104 boys, 77 girls, 61 firstborns) and their mothers. The sample included 87 healthy full-term (born with >37 gestation weeks), 38 low-moderate preterm (born between 32 and 36 gestation weeks), and 56 infants very-extreme preterm (born with <31 gestation weeks). The newborns’ inclusion criteria were (i) be free of sensory or neuromotor impairments, and (ii) be free of severe congenital anomalies and health conditions (e.g., chronic heart disease). 

Data regarding the pre-, peri-, and postnatal characteristics of the infants were collected from hospital charts. A follow-up of this information was collected at 3, 9, and 12 months of corrected age. Parents reported how often their children were ill and how many times they took antibiotics. Only 84 parents were able to state how many times their children were ill (without reasonable doubt). Table 1 informs descriptive statistics of the perinatal and demographic characteristics of the participants.

### 2.2. Procedures and Measurements

We conducted this study following the ethical guidelines outlined in the Declaration of Helsinki. We obtained informed consent from participants or their legal guardians before data collection. Portuguese hospital ethics committees where we performed the study and the Portuguese Data Protection Commission approved the study aims, methods, and ethical procedures.

Two female researchers recruited participants after two days of birth in the case of full-term infants and after five-to-seven days in the case of very-to-extreme preterm infants. Follow-up visits to the laboratory were performed at 9 and 12 months postpartum (corrected age). During the visit at 9-months, mother–infant dyads were videotaped during free-play exchanges. At 12 months, dyads participated in the Strange Situation laboratory procedure [38].

### 2.3. Measures

#### 2.3.1. Mother and Infant Interactive Behavior at 9 Months of Age

We scored the mother–infant interactive behavior in free-play with the Child–Adult Relationship Experimental Index (CARE-Index) [65] and proceeded as in [4]. 

Two trained and blind (against the study) coders on the CARE-Index scored the videotaped free-play interactions. The intercoder reliability evaluated according to [66] and as explained in [4] had an overall average ICCs of 0.73 for infants born preterm [4].

#### 2.3.2. Attachment Patterns

Strange Situation is a laboratory paradigm designed to elicit infant attachment patterns [38]. This experimental situation lasts 21 min in a sequence of seven 3-min episodes. Episodes are interrupted when infants are distressed. The experience increases stress by introducing the infant to an unknown room, interacting with an unfamiliar adult (the stranger), and two separations followed by two reunions with the mother. 

The videos of the Strange Situation were coded in two steps. First, two trained, reliable coders classified the videos as either securely attached (B), insecure-avoidant (A), or insecure-ambivalent (C). This first coding followed the procedures developed by Ainsworth et al. [38]. Next, after describing the infant’s fearful, disorganized, or disoriented behaviors according to the Main and Solomon guidelines for disorganized or disoriented behaviors [47,48], a second classification of mother–infant attachment was performed to describe infant disorganized attachment (D). Cohen’s kappa coefficients indicated good to excellent intercoder reliability.

### 2.4. Analytic Plan

To decide on the analysis to perform, the normality of the variables was tested using Kolmogorov–Smirnov statistics. Next, we conducted three sets of statistical analyses to address the study’s aims. First, we used univariate frequency analysis to obtain the distribution of attachment patterns. Second, we ran cross-tabulations to examine the prevalence of antibiotic use according to each attachment pattern. Chi-square analyses were used to test differences in the prevalence of antibiotic use by attachment pattern. We compared each possible pair of column proportions using z-tests. In addition, we used Bonferroni test correction as a way of proportional reduction in error type I to predict patterns of attachment based on prematurity status. We calculated Cramér’s V to measure how strongly antibiotic use and attachment patterns are associated (effect size). Third, one-way ANOVA was used to evaluate the association between maternal/infant interactive behavior and demographic variables according to the three attachment patterns. Tukey’s post hoc tests were used to test differences between specific attachment groups in one-way ANOVA.

## 3. Results

### 3.1. Associations between Attachment Patterns and Antibiotic Use

Secure attachment prevailed in this sample (72/181; 39.8%), followed by avoidant attachment (54/181; 29.8%), and by ambivalent attachment (37/181; 20.4%). A small group of infants (18/181; 9.9%) presented signals of a high-risk attachment. Under the original attachment scales, we coded these 18 infants as follows: 6 as secure, 5 insecure-ambivalent and 7 insecure-avoidant.

According to Table 2, infant attachment pattern is associated with antibiotic use [χ^2^ (2) = 14.507, *p* < 0.005]. The proportion of ambivalent-attached infants that used antibiotics (64.9%) is higher than among secure (30.6%). The proportion of disorganized infants that used antibiotics was also high (61.1%), although the difference is not statistically significant. For the association between attachment pattern and antibiotic use, there are middle-size effects according to Cramer’s V [ϕ_C_ = 0.283, *p* < 0.005]. Among the six disorganized infants coded as secure under the original attachment scale, two took antibiotics in the first nine months; among the five disorganized infants coded as insecure-ambivalent under the original scale, five took antibiotics; finally, among the seven disorganized infants coded as insecure-avoidant under the original scale, four took antibiotics.

Regarding prematurity, as expected, we found more disorganized attachment in infants born preterm, but differences are not significant [Full-Term: 4 (4.6%), Low-to-Moderate Preterm: 6 (15.8%); Very-to-Extreme Preterm: 8 (14.3%)].

### 3.2. Factors Associated with Attachment Patterns

Infant gestational weight, number of years of maternal formal education, and number of infants’ diseases in the first nine months of life (according to the mothers’ reports) varied according to the infant’s attachment strategy (Table 3). Mothers of infants with disorganized attachment have more years of formal education (*M* = 15.56; *SD* = 2.55) than mothers of infants with a secure attachment (*M* = 12.54; *SD* = 3.74) and mothers of infants with an ambivalent attachment (*M* = 13.38; *SD* = 3.67) [*F*(2) = 3.78; *p* = 0.01]. Furthermore, mothers of infants with disorganized attachment reported fewer diseases (*M* = 1.56; *SD* = 1.88) than other mothers, particularly mothers of infants with avoidant attachment (avoidant infants: *M* = 9.87; *SD* = 9.53; secure infants: *M* = 4.94; *SD* = 8.49; ambivalent infants: *M* = 3.94; *SD* = 5.92) [*F*(2) = 3.38; *p* = 0.03].

Mothers of disorganized-attached infants reported that 61% of their children used antibiotics in the first nine months and that their children were ill 1.56 times on average, while the other mothers (together) reported that only 54% of their children used antibiotics in the first nine months, but their children were sick 3 times more often (5.73 times on average).

According to Table 4, infants securely attached have mothers more sensitive than other infants. In turn, they are less difficult and more cooperative with mothers during free-play interactions than infants with other attachment patterns. Infants with disorganized attachment have mothers less sensitive and are less cooperative than other infants. Moreover, mothers of infants with disorganized attachment presented a more controlling and intrusive behavior than mothers of securely attached infants.

## 4. Discussion

Infants with a disorganized attachment (D) are at risk for social and emotional developmental problems [67]. An earlier study has shown that almost 90% of insecure-ambivalent attached infants took antibiotics in the first nine months of age, but only 32.5% of avoidant infants and 21.5% of secure infants took antibiotics in the same period [1]. The authors [68] speculated that the parents exacerbated worries being reported to the health professionals may raise the professionals’ concerns, leading to preventive prescriptions of antibiotics. Mothers have false conceptions regarding antibiotic use and believe that ‘just in case’, their children should take antibiotics when having an infection, even non-bacterial [69]. Supporting these hypotheses, higher maternal sensitivity was associated with less antibiotic consumption [1]. 

In this study, we applied the D attachment classification to a sample we previously coded with original attachment classification to learn if the amount of antibiotic use is higher in a group of infants generally associated with behavioral perturbation and atypical caregiving [54,70]. Interestingly, we found a similar prevalence of antibiotic use in infants with ambivalent attachment and infants with disorganized attachment (61.1% and 64.9%, respectively) compared with 51.9% of infants with avoidant attachment and 30.6% of securely attached infants. One possible explanation is that the infants coded as disorganized attached were coded as ambivalent-resistant in the first coding using the original attachment scales. But that is not the case; the classification of the 18 disorganized-attached infants in the original scale was: 6 secure, 5 insecure-ambivalent, and 7 insecure-avoidant.

Moreover, although mothers of infants with ambivalent and disorganized attachment reported similar percentages of antibiotic use, ambivalent-attached infants were ill more than twice the number of disorganized-attached infants (on average and per infant: 3.94 times versus 1.56 times, respectively). This mismatch between the number of diseases and the percentage of antibiotic use is even more striking by comparing ambivalent-attached and disorganized-attached infants with the other infants. Although mothers of infants with avoidant and secure attachment reported lower proportions of antibiotic use than mothers of ambivalent and disorganized infants, their children were ill 9.87 and 4.94 times, respectively (hence, more often than ambivalent-attached and disorganized-attached infants).

Concerning the D group, one wonders if the mothers’ reports are reliable or if their infants had antibiotic prescriptions each time they were ill. It is important to stress that mothers of infants classified with a disorganized pattern had higher levels of education than other mothers. Thus, they are able to understand health information and report their infant’s health history. Further investigation of infants classified with disorganized attachment is necessary to understand the quality of health care provided to them and their mother’s mental health (e.g., depression). Moreover, future studies need to observe maternal behavior in daily routines to understand if these mothers are absent and only attend to their infants’ health needs when they are already in pain or sick for a while and already in need of an antibiotic prescription. 

To illustrate the major findings of this study, Figure 1 compares the proportions of antibiotic use, number of diseases, and maternal sensitivity of the four attachment patterns. Each value (each column) represents the ratio of the mean value for each variable and for each pattern with the maximum value of that variable. For example, because the mean value of the maternal sensitivity of mothers of infants with disorganized attachment is 4.72 and the maximum value in that variable is 8.64 (mothers of secure-attached infants), the corresponding column in the disorganized-attached infants is at the value 4.72/8.64 = 0.55, and the column of secure-attached infants is at 8.64/8.64 = 1 (Figure 1).

As prior research documented, mothers of infants classified with disorganized patterns were less sensitive to their infant’s needs and more controlling or unresponsive than mothers of other infants [56]. Moreover, the mothers of infants with disorganized attachment had very low means of maternal sensitivity, indicating high-risk interactions according to the CARE-Index manual and comparing with the mean values of past Portuguese studies on maternal sensitivity [71]. These results add to the body of research that indicates that the relationships of infants with disorganized attachment are atypical and high-risk. However, it is essential to acknowledge that environmental factors such as caregiving are not the only ones contributing to these infants’ behavioral perturbation. Genetic factors may also play an important role. Indeed, a polymorphism of a gene (DRD4, dopamine receptor D4 gene) is associated with disorganized attachment (DRD4 7-repeat polymorphism) [72]. However, there are significant environment×genetic factors interactions [58,73]: while the 7-repeat allele increased the levels of disorganized attachment if maternal sensitivity is low (high-risk environment), the same allele was present in infants with the lowest levels of disorganized behavior if the mother was sensitive (low-risk environment). Supporting the thesis that disorganized attachment results from biological disposal and environmental influences (gene×environment interactions), one study found an association between another gene polymorphism and attachment disorganization: the short allele variation of the 5-HTT gene (encoding the serotonin transporter protein) was associated with the disorganized pattern when maternal responsiveness was low, but not when responsiveness was high [74].

One cannot change infants’ genomes (or, at least, it would be an arduous and costly task). However, these studies show that infants with disorganized attachment have mothers with very low levels of sensitivity. Family-centered early intervention may prevent these inadequate interactions. For that purpose, the support for parents should start as early as during pregnancy, including screening maternal and paternal mental health in obstetric care. Cases of mental problems should be immediately addressed. The transition from the maternity ward to the family’s home (postpartum discharge) is another key moment. Some parents may struggle with the transition to parenthood once leaving a structured clinical environment and without health professionals; it is necessary to support parents in adapting to this period [75]. Attachment strategies are not deterministic and do not correspond to fixed personality traits. These strategies result from a relational process in which children learn their worth or value, whether they can trust their attachment figures, and if they are loved [76]. These three notions are central to developing representations of relationships [77]. Therefore, it is possible to alter these representations through a family-centered intervention, reinforcing the strengths of families and repairing parental traumas [78,79].

Overall, this research sheds light on the complex interplay between psychological processes, early relationships, and health outcomes, calling for further investigation and targeted interventions to mitigate the adverse effects of disorganized attachment on children’s rights and well-being.

## 5. Conclusions

In conclusion, this study provides evidence of the association between attachment relationships, antibiotic consumption, and maternal sensitivity in infants with disorganized attachment.

Our findings revealed that 61.1% of children exhibiting disorganized attachment had taken antibiotics, yet they experienced sickness less often than the other children. This result suggests these children are significantly more inclined to use antibiotics concerning the number of illnesses they encounter, compared to children displaying different attachment patterns.

Moreover, mothers of children with disorganized attachment showed very low levels of maternal sensitivity, indicating high-risk interactions. Strikingly, these mothers had high levels of literacy. Future studies need to include maternal stress and mental health variables to understand the behavior of mothers of disorganized-attached infants.

The findings emphasize the need for early intervention and support for mothers and infants with disorganized attachment to promote healthier attachment relationships and potentially reduce antibiotic use in infancy.

## Figures and Tables

**Figure 1 children-10-01232-f001:**
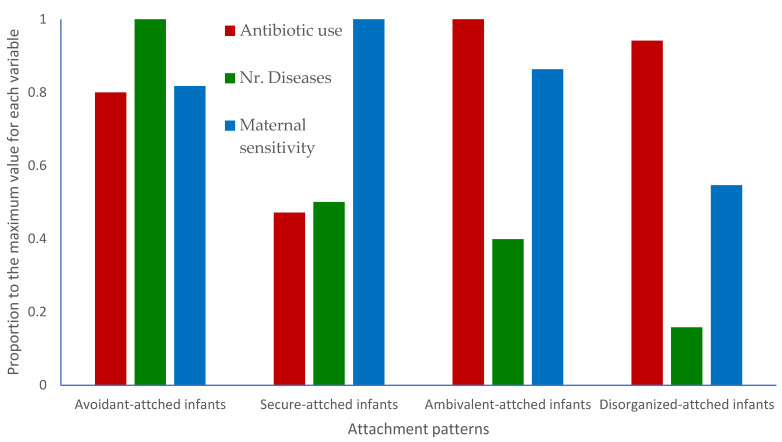
Differences between the four attachment patterns. Each column represents the ratio Mean/Maximum(Means across patterns) of the mean value for each variable and for each pattern with the maximum value of that variable. For example, because the mean value of the number of diseases of disorganized-attached infants is 1.56 and the maximum value in that variable is 9.87 (avoidant-attached infants), the corresponding column in the disorganized-attached infants is at the value 1.56/9.87 = 0.16 and the column of avoidant-attached infants is at 9.87/9.87 = 1.

**Table 1 children-10-01232-t001:** Samples’ characteristics.

	*M*	*SD*	Min.	Max.
Gestational age (weeks)	34.9	4.33	23.5	41.3
Gestational weight (kg)	2385	1026.0	500	4635
Apgar at first minute	7.98	1.761	1	10
Apgar at fifth minute	9.31	1.058	2	10
Maternal age (years)	30.9	5.52	18	46
Maternal education (years)	13.5	3.79	3	23
Number of siblings	0.91	0.884	0	5
Number of diseases	5.7	8.29	0	28

**Table 2 children-10-01232-t002:** Crosstab of attachment patterns and antibiotic use.

	Attachment Classification 12 Months *			
Antibiotic	Avoidant	Secure	Ambivalent	Disorganized
Yes	28 (51.9%, 0.9) ^a,b^	22 (30.6%, −3.6) ^a^	24 (64.9%, 2.4) ^b^	11 (61.1%, 1.3) ^a,b^
No	26 (48.1%, −0.9) ^a,b^	50 (69.4%, 3.6) ^a^	13 (35.1%, −2.4) ^b^	7 (38.9%, −1.3) ^a,b^

* Each subscript letter denotes a subset of antibiotic categories (yes or no) whose column proportions do not differ significantly from each other at the 0.05 level.

**Table 3 children-10-01232-t003:** ANOVA-test for mean differences of infant and maternal factors according to attachment patterns.

	Insecure-Avoidant *M (SD)*	Secure *M (SD)*	Insecure-Ambivalent *M (SD)*	Disorganized *M (SD)*	*F*	*p*	Tukey HSD	*η* ^2^
**Maternal variables**								
Maternal age	31.28 (5.00)	30.63 (5.43)	31.86 (5.96)	29.28 (6.32)	1.04	0.38	-	0.02
Maternal formal education	14.00 (3.99) ^a^	12.54 (3.74) ^b^	13.38 (3.67) ^c^	15.56 (2.55) ^d^	3.78	0.01	d > b,c	0.06
**Infant interactive behavior**								
Number of siblings	0.98 (0.98)	1.06 (0.87)	0.68 (0.71)	0.61(.85)	2.38	0.07	-	0.04
Gestational age	34.57 (4.76)	35.93 (3.97)	34.10 (4.11)	33.16 (4.05)	2.94	0.04	*	0.05
Gestational weight	2340 (1101) ^a^	2650 (990) ^b^	2073 (933) ^c^	2098 (904) ^d^	3.38	0.02	b > a	0.05
Apgar 1	8.06 (1.76)	7.96 (1.86)	8.03 (1.50)	7.72 (1.96)	0.17	0.92	-	0.00
Apgar 5	9.39 (0.83)	9.30 (1.30)	9.32 (0.82)	9.06 (1.06)	0.45	0.72	-	0.01
Number of diseases	9.87 (9.53) ^a^	4.94 (8.49) ^b^	3.94 (5.92) ^c^	1.56 (1.88) ^d^	3.28	0.03	a > d	0.11

* The Tukey HSD analysis to compare columns showed no differences group by group for gestational age. Each superscript letter denotes a subset of attachment at 12-months categories whose column proportions is tested with Tukey HSD test (denoted with normal letters: a, b, c and d).

**Table 4 children-10-01232-t004:** ANOVA-test for mean differences of infant and maternal interactive behavior according to attachment patterns.

	Insecure-Avoidant *M (SD)*	Secure *M (SD)*	Insecure- Ambivalent *M (SD)*	Disorganized *M (SD)*	*F*	*p*	Tukey HSD	*η* ^2^
**Maternal interactive behavior**								
Sensitivity	7.06 (1.89) ^a^	8.64 (2.14) ^b^	7.46 (2.40) ^c^	4.72 (1.90) ^d^	18.26	0.00	b > a,c,d & a,c > d	0.236
Control/ Intrusivity	4.50 (3.32) ^a^	3.31 (2.90) ^b^	3.59 (3.17) ^c^	5.89 (3.50) ^d^	3.99	0.01	b < d	0.063
Unresponsivity	2.35 (3.02) ^a^	2.14 (2.53) ^b^	2.86 (2.99) ^c^	3.22 (2.92) ^d^	1.033	0.38	b > a,c & c > a	0.017
**Infant interactive behavior**								
Cooperativity	7.07 (2.08) ^a^	8.61 (1.92) ^b^	7.43 (2.56) ^c^	5.00 (1.97) ^d^	15.71	0.00	b > a,c, & a,c > d	0.210
Compulsivity/Compliance	2.70 (3.51)	1.50 (2.82)	1.81 (2.74)	3.50 (4.20)	2.769	0.04	*	0.045
Difficulty	3.54 (3.47) ^a^	2.03 (2.43) ^b^	3.35 (3.12) ^c^	4.50 (3.90) ^d^	4.557	0.00	b < a,c,d	0.062
Passivity	1.04 (1.81)	1.79 (1.97)	1.51 (1.87)	1.11 (2.19)	1.757	0.16	-	0.029

* The Tukey HSD analysis to compare columns showed no differences group by group for compulsivity/compliance. Each superscript letter denotes a subset of attachment at 12-months categories whose column proportions is tested with Tukey HSD test (denoted with normal letters: a, b, c and d).

## Data Availability

The data presented in this study are available on request from the corresponding author. The data are not publicly available due to data confidentiality and to protect participants’ privacy.

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
