# Peer review of "Antibiotic Consumption, Illness, and Maternal Sensitivity in Infants with a Disorganized Attachment"

_children, 2023, doi:10.3390/children10071232_

Round 1

Reviewer 1 Report

the introduction is clear and detailed. The bibliographic references must be numbered and in parentheses, in some cases this is not the case see line 84, 109,  116, going on you will find other  incorrect bibliographic references. The goal is missing because this work is important. 

The number of partecipants dicrete. Good, measures used for free play interaction between mother and child 

Clear and interesting results 

Author Response

We thank the reviewer for the comments.

We corrected the references. In some cases, we write the authors and the reference in the correct format appears at the end of the sentence. For example, in line 86 we write "The study by Murray et al. also estimated that about 5 million deaths in 2019 were associated with antibiotic resistance [11]. " 

Thank you for your nice words.

Reviewer 2 Report

I must admit that this is my first time reading about this topic.

I am very interested in antibiotic use and the type of attachment.

I am very pleased that you offered infoabout the preterm babies, as I am a neonatologist.

In my opinion, there might be other factors that are interfering with the type of attachment, but in this article, the evidence you provided is good enough.

Your paper meets the criteria for being published in this journal.

English is fine

Author Response

We thank the reviewer for the comments.

We revised the English and ameliorated the introduction as requested. We also increased the number of references in the introduction.

Best regards,

Francisco Dionisio and co-authors
